# Transcriptomic profiling of human cardiac cells predicts protein kinase inhibitor-associated cardiotoxicity

J. G. Coen van Hasselt [1,2,7], Rayees Rahman[1,7], Jens Hansen [1], Alan Stern[1], Jaehee V. Shim[1], Yuguang Xiong[1], Amanda Pickard[1], Gomathi Jayaraman[1], Bin Hu[1], Milind Mahajan[3], James M. Gallo[1,4], Joseph Goldfarb[1], Eric A. Sobie[1], Marc R. Birtwistle [1,5], Avner Schlessinger [1,8✉], Evren U. Azeloglu [1,6,8✉] & Ravi Iyengar [1,8✉]

Kinase inhibitors (KIs) represent an important class of anti-cancer drugs. Although cardiotoxicity is a serious adverse event associated with several KIs, the reasons remain poorly understood, and its prediction remains challenging. We obtain transcriptional profiles of human heart-derived primary cardiomyocyte like cell lines treated with a panel of 26 FDA-approved KIs and classify their effects on subcellular pathways and processes. Individual cardiotoxicity patient reports for these KIs, obtained from the FDA Adverse Event Reporting System, are used to compute relative risk scores. These are then combined with the cell line-derived transcriptomic datasets through elastic net regression analysis to identify a gene signature that can predict risk of cardiotoxicity. We also identify relationships between cardiotoxicity risk and structural/binding profiles of individual KIs. We conclude that acute transcriptomic changes in cell-based assays combined with drug substructures are predictive of KI-induced cardiotoxicity risk, and that they can be informative for future drug discovery.

[1] Department of Pharmacological Sciences and Systems Biology Center New York, Icahn School of Medicine at Mount Sinai, New York, NY, USA. [2] Division of Systems Biomedicine and Pharmacology, Leiden Academic Centre for Drug Research, Leiden University, Leiden, Netherlands. [3] Department of Genetics and Genomic Sciences, and Icahn Institute for Genomic Sciences and Multiscale Biology, Icahn School of Medicine at Mount Sinai, New York, NY, USA. [4] Department of Pharmaceutical Sciences, School of Pharmacy and Pharmaceutical Sciences, University at Buffalo, Buffalo, NY, USA. [5] Department of Chemical and Biomolecular Engineering, Clemson University, Clemson, SC, USA. [6] Deparment of Medicine, Division of Nephrology, Icahn School of Medicine at Mount Sinai, New York, NY, USA. [7] These authors contributed equally: J. G. Coen van Hasselt, Rayees Rahman. [8] These authors jointly supervised this work: Avner Schlessinger, Evren U. Azeloglu, Ravi Iyengar. ✉email: avner.schlessinger@mssm.edu; evren.azeloglu@mssm.edu; ravi.iyengar@mssm.edu

Protein kinase inhibitors (KIs) are an important class of therapeutics used for the treatment of various forms of cancer[1,2] and other diseases. There are currently more than 48 KIs approved for clinical use by the U.S. Food and Drug Administration (FDA) and other regulatory agencies[3], and more than 250 KIs are undergoing clinical trials or are in development[4–6]. The clinical effectiveness of KIs as cancer drugs has led to a broad effort to develop drugs that are more efficacious and have reduced the propensity for adverse events. Cardiotoxicity (CT) is a clinically important adverse event associated with several KIs[7–10]. KI-associated CT manifests as loss of cardiomyocyte function, which can lead to heart failure[11]. Given the extensive therapeutic potential of KIs, approaches to identify and subsequently mitigate the risk for CT during early development of novel KIs and during clinical administration are urgently required.

We do not yet sufficiently understand the mechanisms underlying KI-associated CT. The human kinome consists of more than 500 protein kinases[12]. Given that many KIs exhibit multitarget pharmacology[13], inhibition of multiple protein kinases in cardiomyocytes may lead to adverse drug effects such as CT[14]. For individual KIs, pathways involved in mitochondrial function[8,15,16], endoplasmic reticulum stress response[16], and AMPK inhibition[17], have been shown to be associated with KI-induced CT[18]. Overall, however, the general mechanisms of KI-induced CT are still poorly understood[18].

Obtaining quantitative clinical risk scores for KI-associated CT is also challenging, as the risk for KI-associated CT has not been systematically studied. The FDA adverse event report system (FAERS) database has been previously applied to quantify the risk of ADRs[19–21]. The FAERS database contains over 9 million individual drug-associated adverse-event reports reported by industry and physicians. Through statistical analyses of the FAERS database, relatively unbiased estimates for the relative risk for specific ADRs can be computed. Such risk scores are clinically relevant as they are based on real-life patient population, and they are not solely based on selected patient cohorts. We previously used such analyses of the FAERS database in combination with systems' pharmacology-based approaches to obtain mechanistic insights into adverse-event mechanisms[21,22].

In the current study, generated as part of the NIH-funded Library of Integrated Network Based Cellular Signatures (LINCS) Drug Toxicity Signature Generation Center (DToxS), we take a top–down global approach to determine if a comprehensive profiling of gene expression changes in human cardiomyocytes can provide insight into pathways associated with KI-induced CT, and to potentially predict the risk of CT. The rationale for this approach is based on the central assumption that CT largely originates from cardiomyocytes where one or more protein kinases contribute to the pathophysiology. Since progression to heart failure takes several months to manifest, it is not immediately obvious if gene expression changes measured after drug treatment for a few days would have predictive value. Thus, a second important assumption is that early changes in gene expression upon drug treatment of cardiomyocytes are indicative of later physiological events. We test the validity of our assumptions by experimentally obtaining gene-expression patterns for the different KIs, and if these patterns could be selectively associated with the clinical risk of CT for each KI, thereby providing gene-expression signatures for KI-associated CT.

We report the generation of transcriptomic profiles from four human primary cardiomyocyte-like cell lines. These profiles are generated using 23 KIs that were FDA-approved and used extensively at the time of experimental design, such that an adequate number of clinical reports have been collected. Drugs are used at their imputed therapeutic concentrations. Through this pan-KI transcriptomic profiling, we obtained insights into the affected pathways that may be related to KI-associated CT. We show that selective patterns of gene expression can be associated with the FAERS-derived clinical risk for KI-associated CT, which may be highly relevant to identify KI drug candidates at risk for showing clinical CT. We also describe the relationships between KI CT risk and structural properties of KIs, highlighting the potential for re-engineering small molecules that exhibit a high risk for CT.

## Results

**Differences in CT risk of kinase inhibitors**. In order to obtain unbiased estimates of clinical risk of KI-associated CT, we analyzed individual adverse-event reporting data from FAERS (Fig. 1a). Reporting odds ratios (RORs) were derived based on the relative frequencies of AE occurrence of each KI compared to all KIs. These risk scores provide a relative ranking of KI-associated toxicity. Kinase inhibitors were shown to have pronounced differences in the relative risk of CT (Fig. 1b). When comparing the ranking of risk scores derived from FAERS with adverse drug-reaction (ADR) reporting data from the World Health Organization (WHO) ADR reporting database, we find that the ranking from these databases largely agrees (Fig. 1c), indicating the general consistency of the clinical risk scores across databases.

**Phenotypic assays poorly correlate with CT**. We performed a literature review for in vitro and in vivo experimental datasets that aimed to predict CT risk based on phenotypic readouts, such as cell viability or beating rate from in vitro cardiomyocyte or animal models, to determine if such phenotypic experiments can predict the clinical risk scores for CT. Studies in which drugs at the clinical concentration induced more than a 20% change in various phenotypic readouts compared to control experiments were classified as predicting potential CT (Fig. 1d). Across these studies, it was apparent that there was no identifiable relationship between apparent experimental toxicity in comparison to the relative incidence of CT in patients as derived from FAERS.

We conducted dose–response experiments with selected KIs that had varying risks for CT using the cardiomyocyte cell lines that were used in the current study for transcriptomic profiling, quantifying cell viability, and mitochondrial stress after 48 h of exposure to the selected KIs. We again assessed if drugs caused more than a 20% change in cell viability and mitochondrial stress at the typical clinically used concentration (Supplementary Table 1). These studies showed a similar lack of correlation with clinical risk (Fig. 1e, Supplementary Fig. 1). These findings underscore the need for alternative approaches such as early molecular signatures for CT. This identified lack of the predictiveness of preclinical in vitro and in vivo phenotypic assays, as has been noted by others[7].

**Transcriptomic profiling of human primary cardiomyocyte-like cell lines**. To study the transcriptomic response to KIs associated with CT, we obtained four primary cardiomyocyte lines that were isolated from ventricles of healthy adult human heart (two male and two female, PromoCell GmbH, Germany). Culture conditions, detailed phenotypic characterization of each cell line, including gene and protein expression, morphology, and functional assays, can be found on the DToxS Center website (www.dtoxs.org) under the "Cellular Metadata" section.

Confluent cardiomyocyte-like cells were treated with drugs for 48 h at concentrations similar to their clinical concentration (Supplementary Table 1) with 3–4 replicates and 3–4 cell lines (Supplementary Table 2), after which RNA was extracted and sequenced using the 3′ digital gene-expression method[23] (Fig. 2a).

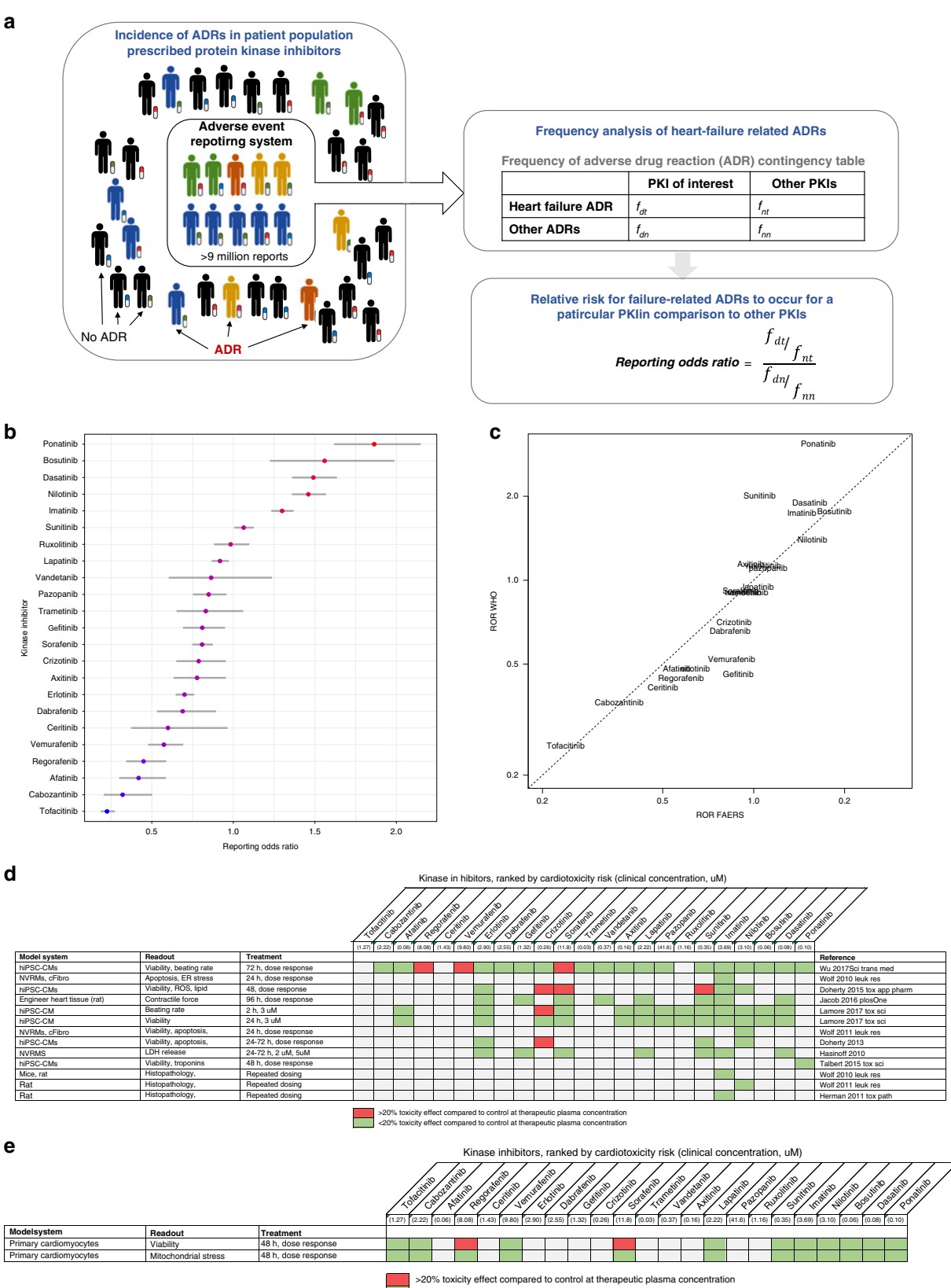

**Fig. 1 Cardiotoxicity of protein kinase inhibitors. a** Approach to quantify relative clinical cardiotoxicity risk scores for kinase inhibitors from the FDA Adverse Event Reporting System (FAERS) database. **b** Reporting odds ratio (mean and 95% confidence interval of computed odds ratio) for cardiotoxicity across kinase inhibitors from FAERS. **c** Comparison of ranking derived from FAERS and WHO Pharmacovigilance data shows agreement. **d** Literature-reported in vitro and in vivo preclinical assays to predict KI-associated cardiotoxicity poorly correlated with clinical FAERS-derived risk scores for cardiotoxicity at clinical drug concentrations. **e** In vitro dose–response experiments for selected KIs for viability and mitochondrial stress poorly correlate with clinical FAERS-derived risk scores for cardiotoxicity. Source data are provided in source data file.

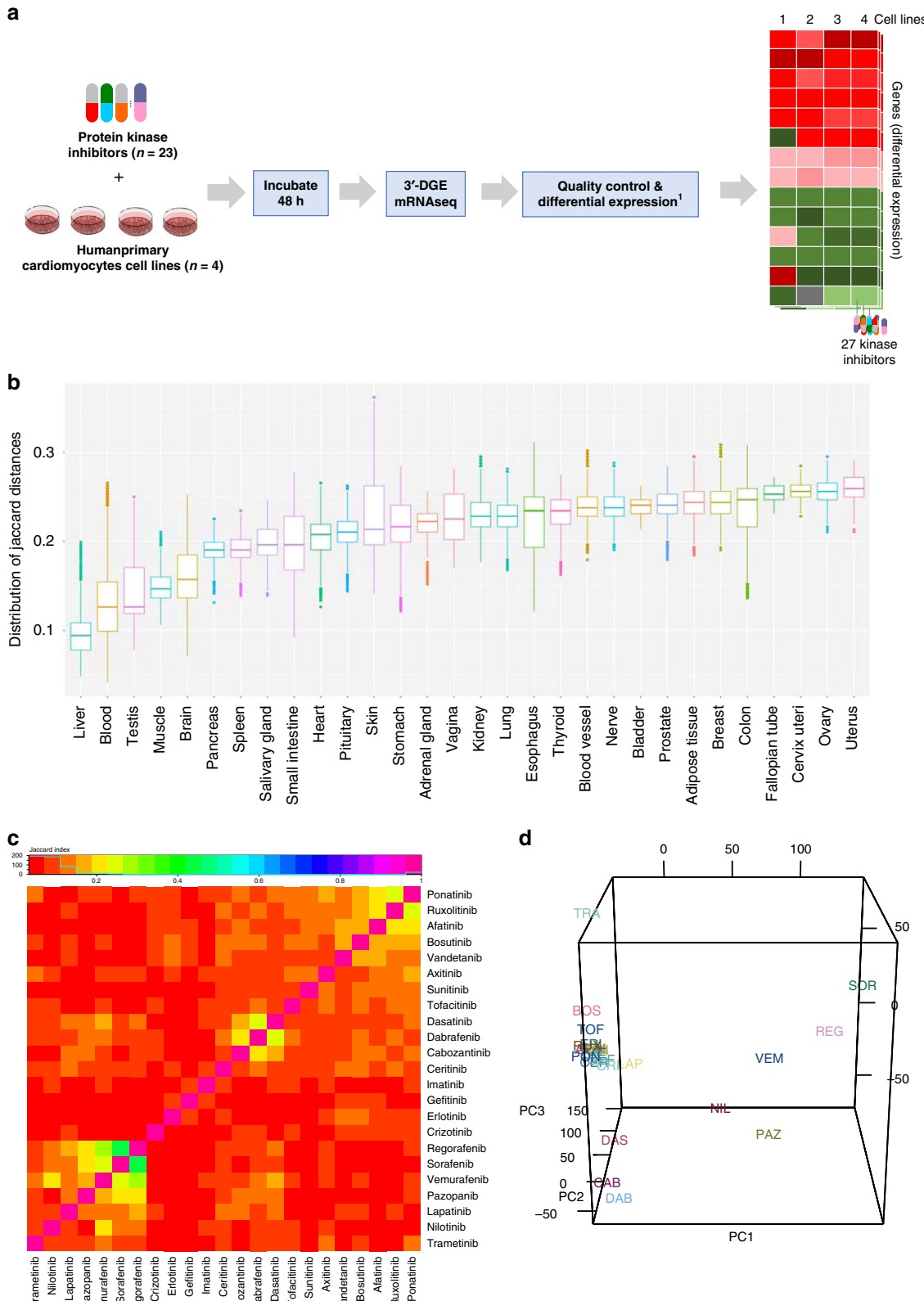

We investigated if transcriptomic profiles of PromoCell cardiomyocytes are related to human heart tissue and hence a good model to study CT. We compared the gene-expression similarity of untreated PromoCell cardiomyocytes against tissues available in the Genotype-Tissue Expression (GTEx) project, which contains gene-expression data from many human tissues, including the heart (Fig. 2b)[24]. Using the Jaccard distance for the top expressed 250 genes (based on transcript per million counts) for both untreated PromoCell and GTEx tissues, we observe that PromoCell cardiomyocytes' expression exhibits a gene expression similar to blood (rank 2), muscle (rank 4), and heart (rank 10) tissue. Based on these results, we conclude that the PromoCell

**Fig. 2 Overview of pan-KI transcriptomic profiling in human primary cardiomyocyte-like cells. a** Overview of experimental approach to generate transcriptomic data. For each drug, genes were ranked by absolute mean fold-change gene-expression value across replicates (>3 biological replicates) and cell lines (a total of 1309 experiments), and the top 250 genes for each KI were kept. Information about the total number of replicates can be found in the source data file. **b** Jaccard similarity of gene-expression signature of PromoCell cardiomyocyte cell lines (102 samples) to gene-expression signatures of tissues available in the GTeX database (17,382 total samples). Boxplot whiskers refer to the upper and lower quartile of all pairwise Jaccard coefficients between each sample, within each tissue type. Information about each boxplot's sample size, minima, maxima, and center is provided in the source data file. **c** Heatmap depicting the Jaccard index that indicates the magnitude of similarity in top-ranking differentially expressed genes for all KI pairs. **d** First three principal components (PCs) based on full mean fold-change gene-expression profiles across KIs. Source data are provided in source data file.

cardiomyocytes can offer comparable gene-expression changes to that of cardiomyocytes.

**Limited overlap in differentially expressed genes across KIs.** Differential gene-expression fold-change values were computed across the four cell lines. Initial analyses showed that the DEGs generally clustered more strongly by drugs than by cells. We calculated median fold-change values for each KI across cell lines, resulting in a single gene-expression profile for each KI. Ranked gene lists for each KI were generated by ranking by differential gene-expression $p$ value and keeping the top 250 genes. To assess the similarity between genes present in the top 250 genes for each KI, the Jaccard index was calculated for each ranked list of KI-specific genes, which indicated a limited overlap (<0.25) between the top 250 genes across KIs (Fig. 2c). Principal component analysis showed variable gene-expression patterns for nine KIs, while for the remaining KIs, little variation in gene expression was seen (Fig. 2d), even though these remaining KIs included drugs for which CT is well established. We concluded that ranked differential gene-expression values would not be sufficient to provide clear insights into gene-expression profiles associated with CT.

**Pathways correlated with KI-associated CT.** To identify pathways and subcellular processes across KIs and their potential involvement with CT, we performed enrichment analysis for protein kinases and KEGG terms using the top 250 differentially expressed genes ranked by $p$ value across cell lines and KIs. We then correlated $p$ values of enriched terms with clinical FAERS-derived risk scores to identify potential kinases and pathways associated with CT risk (Fig. 3a). The protein kinase LIMK2, which is involved in actin cytoskeleton reorganization pathways, ranked the highest in its correlation specifically enriched for KIs with a higher risk score (Fig. 3b). Sucrose- and pyruvate-metabolism pathways were the most strongly enriched pathways correlating with high risk scores (Fig. 3c). However, since no directionality in pathways is considered in these enrichment analyses, both the positively and negatively correlated processes may play a role in the development of CT. When considering enriched protein kinases and KEGG processes across all KIs without considering correlation to CT risk, multiple pathways were identified (Supplementary Fig. 2). These findings indicate that there is likely substantial complexity underlying the action of KI in cardiomyocytes, although currently these analyses remain correlational and do not offer proof of causal relationships.

**Transcriptomic signature to predict CT risk.** We tested if our KI-wide fold-change gene-expression profiles correlated with the KI-specific clinical risk scores for CT to identify a predictive transcriptomic signature for CT risk. Given the limited similarity between top-ranking gene-expression profiles across KIs, the entirety of the gene-expression profiles for different KIs were considered as potential predictors for KI-associated CT risk. KI-specific expression profiles of 10,749 genes were available as potential predictors for KI-specific CT risk scores. To identify

genes most strongly associated with CT risk, we used an elastic net-penalized regression approach, which aims to select the most predictive variables while avoiding overfitting[25].

A two-stage regression analysis was performed (Fig. 4a). From the available 23 KIs with the associated clinical CT risk scores, we randomly left out 2 KIs for external validation of the model (test set, 10% of data). The differential gene-expression profiles of 21 remaining KIs were then used to train the model. Given the limited number of available drugs, small changes in expression patterns for drug were expected to affect the identity of the overall set of predictor genes. Therefore, we generated bootstrap datasets by random resampling of KI risk and the associated gene-expression profiles. These bootstrapped datasets were then fit using elastic net models. This first step was performed to identify gene-based predictors that could consistently predict CT risk and contributed significantly to the prediction of this risk. The bootstrap analysis resulted in stable selection of potential predictors. Predictors to be included in the final elastic net regression model were selected based on their minimal root-mean-squared prediction error (RMSE) after cross-validation. Based on this cross-validation, the gene-expression-based predictors in the final elastic net models consisted of 26 genes with the associated variable importance values (Fig. 4b).

Repeated cross-validation analyses indicated good predictive performance of the model for left-out KIs (Fig. 4c). We evaluated our 26-gene signature for predicting CT risk on an independent validation set of six KIs, of which three KIs were previously untested (Fig. 4d). We note that the independent validation set was performed 1 year after the original signatures were generated, using a different experimental protocol for the transcriptomic assay that was based on mRNA detection using random primers. We observed accurate predictive performance for five out of six KIs tested. The outlier, ibrutinib, had the lowest, albeit acceptable, predictive performance, with an error of 0.493 between the predicted and observed risk scores. Taken together, the developed signature can be of relevance to support risk prioritization of newly developed KIs. When we tested which of the 21 KIs drove the prediction strength of the model, we found that excluding any of four low-CT risk drugs (cabozantinib, tofacitinib, pazopanib, and erlotinib) increased the error substantially, indicating that these KIs contribute distinct information to the signature. In contrast, several of the high-ranking CT drugs could be excluded without sacrificing accuracy (Supplementary Fig. 3).

We then used the 26-gene signature to construct a protein–protein interaction network analysis to identify protein kinases and transcription factors associated with the signature (Supplementary Fig. 4). Several protein kinases were retrieved that are both known targets of the studied KIs, and which may be associated with the occurrence of KI-induced CT.

**Chemical structures of KIs inform CT risk.** Off-target binding or polypharmacology is commonly observed in KIs[23]. Since off-target binding is dependent on the structure of the drug, we investigated the relationship between kinase inhibitor chemical structure, binding target profile, and CT risk. To do this, we

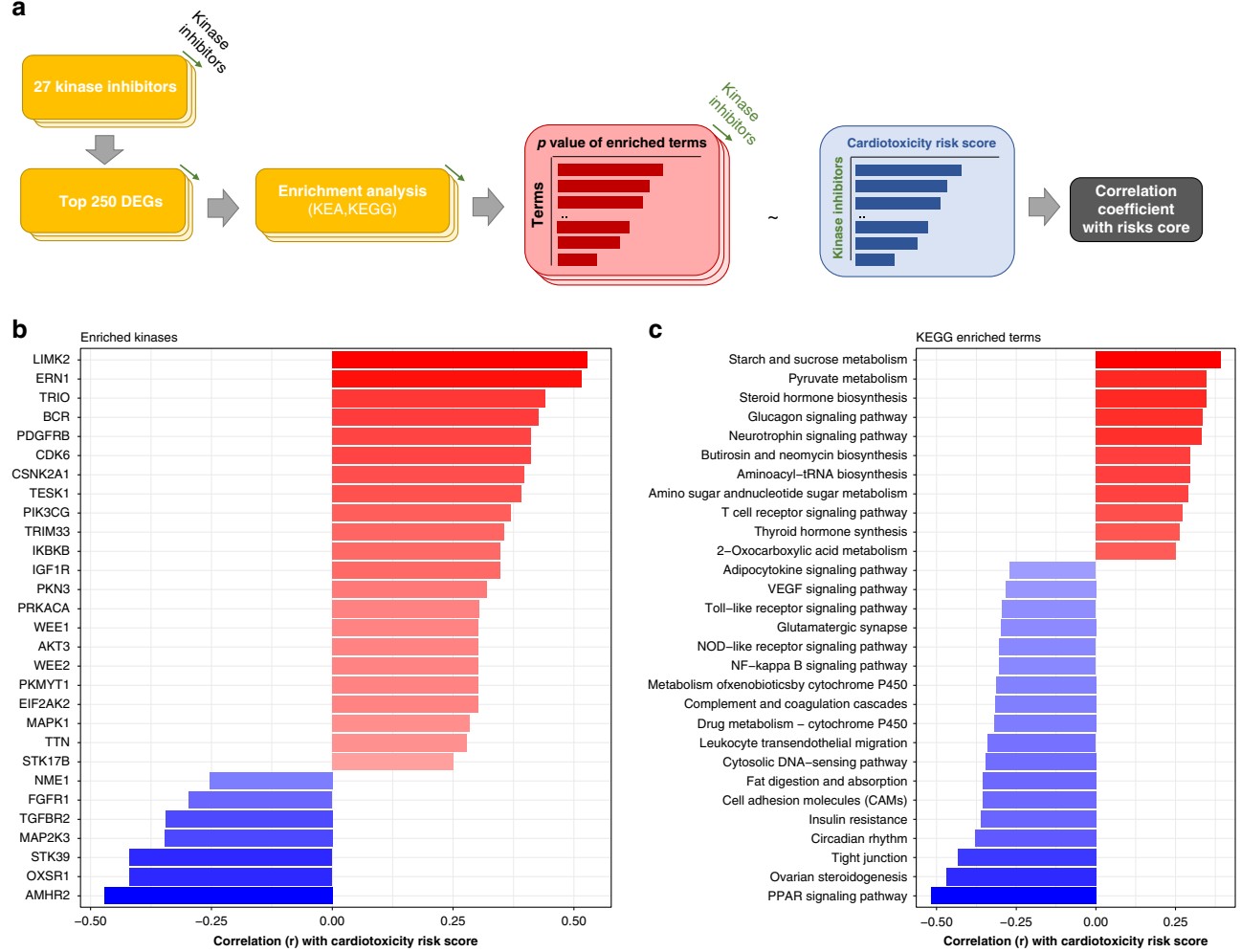

**Fig. 3 Analysis of transcriptomic profiling data in relation to cardiotoxicity risk. a** Flowchart indicating ranked lists of top 250 differentially expressed genes ranked by p value for each kinase inhibitor across cell lines from the transcriptomic cardiomyocyte profiling, which were then enriched and subsequently related to clinical cardiotoxicity risk scores. Enriched kinases (**b**) and enriched KEGG pathways (**c**) ($p < 0.05$) that show a correlation coefficient $> |0.25|$ with cardiotoxicity risk scores and the associated enrichment p values. Source data are provided in source data file.

generated a structure–activity–similarity (SAS) map of the 26 tested inhibitors (in both the training and validation set) and their CT-risk score (Fig. 5A)[26]. SAS maps can be divided into four quadrants: the upper-left quadrant shows KI pairs with low chemical similarity and large changes in CT risk. The lower-left quadrant describes largely dissimilar KI pairs with small changes in CT risk. The lower-right quadrant describes KI pairs that exhibit a "smooth" structure–activity relationship, that is, small changes in chemical similarity are associated with small changes in CT risk. Finally, the upper-right quadrant indicates highly chemically similar compounds with large changes in CT risk.

KI pairs in the upper-right region represent activity cliffs, that is, that small changes in chemical structure are associated with large changes in CT risk. In this region, we find several KI pairs, in particular, we observe large activity cliffs between afatinib and bosutinib as well as bosutinib and erlotinib. Here, all four compounds have the same chemical core (Fig. 5b); however, both afatinib and erlotinib show respectively lower CT risk scores compared to bosutinib. We hypothesized that harmonization of drug substructure, similarity, and promiscuity in the context of kinase inhibitor type may inform on our ability to predict CT risk (Fig. 5c).

By investigating their KI target profiles, we observe that both afatinib and erlotinib are less promiscuous KIs compared to

bosutinib (which is one of the most promiscuous KIs in this set, Fig. 5d), and they both inhibit EGFR at nanomolar concentrations. On the other hand, less promiscuous KIs, such as lapatinib and gefitinib, exhibit a comparably lower CT risk score (Fig. 5e). Indeed, we observe a correlation between kinase inhibitor promiscuity and the observed CT risk score (Supplementary Fig. 5). However, KI promiscuity may not be the sole determinant of CT risk. For example, KIs such as imatinib and nilotinib are not as promiscuous as bosutinib; however, both exhibit relatively high CT risk scores. In this case, both imatinib and nilotinib CT may be explained due to their similar chemical structure and high specificity for protein kinases such as DDR1 and ABL.

Finally, kinase inhibitors have distinct binding modes against their targets[6,27,28]. Kinase inhibitors that bind their kinase targets can be classified based on their binding mode, including the kinase conformation they bind and/or type of interactions they make with their kinase targets (e.g., covalent vs. non-covalent)[6,27,29]. For example, type I inhibitors bind an active kinase conformation, while Type I1/2, II–V bind distinct inactive states (Methods); type VI KI binds the kinase target covalently. We do not observe a clear relationship between kinase inhibitor-binding mode and CT. For example, the type II inhibitors imatinib and nilotinib are observed to have a high CT risk, while the type II inhibitors sorafenib and regorafenib

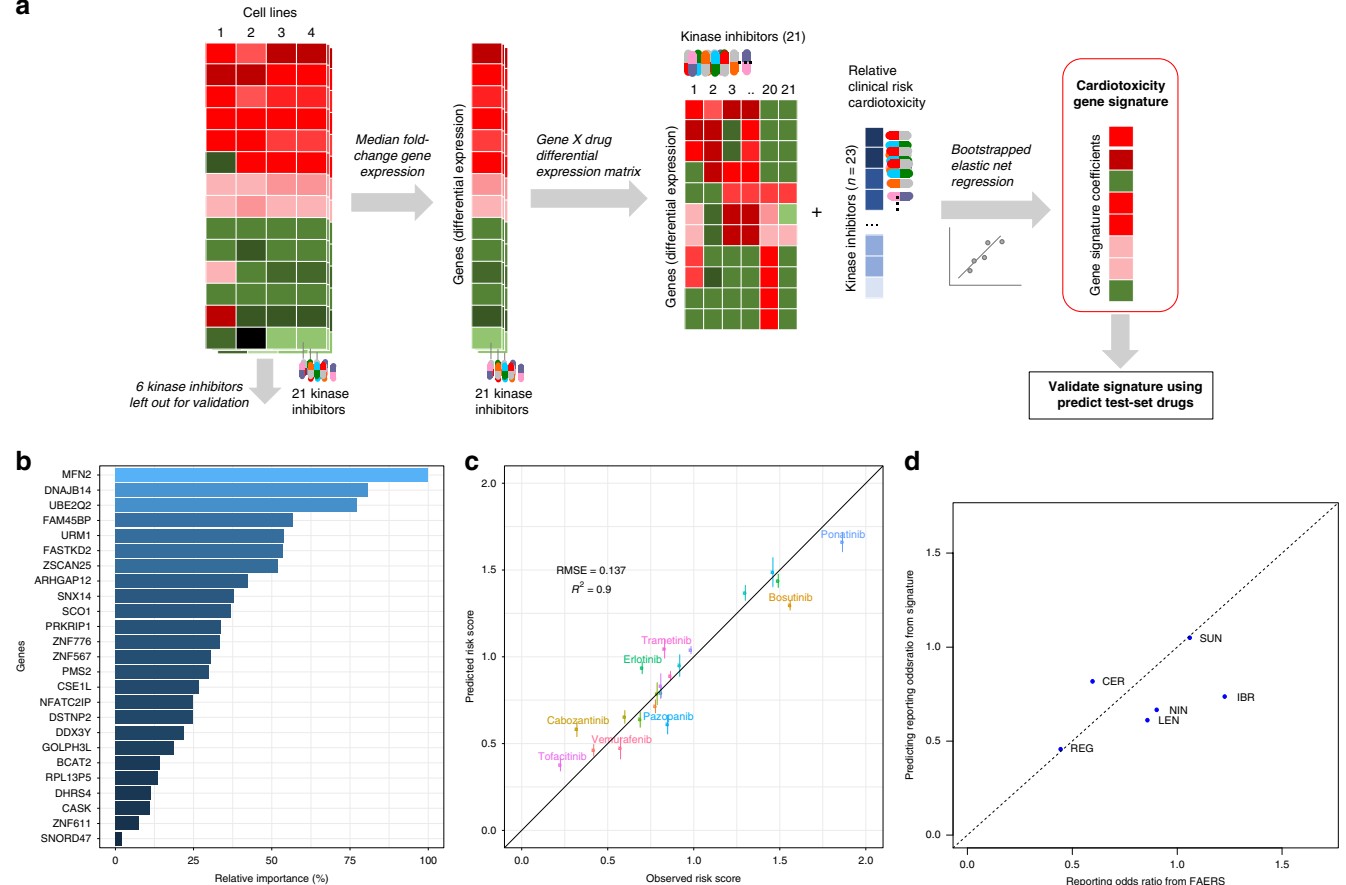

**Fig. 4 Regression analysis for transcriptomic signatures to predict clinical risk. a** Overview of processing and elastic net regression analysis of transcriptomic data in combination with FAERS-derived clinical risk scores. **b** Transcriptomic signature genes selected to predict cardiotoxicity risk score indicating their variable importance. **c** Observed and predicted risk scores from the elastic net cross-validation analysis (mean and standard deviation). **d** External validation of the signature for six kinase inhibitors: regorafenib (REG), sunitinib (SUN), ibrutinib (IBR), lenvatinib (LEN), nintendinib (NIN), and ceritinib (CER).

have comparatively lower observed CT risk. However, both pairs of inhibitors are highly chemically similar and have similar binding targets. Taken together, the observed CT risk of a KI may be related to both a kinase inhibitor's selectivity and its chemical structure. Furthermore, we observe a relationship between chemical structure and binding target similarity to the predictive performance of our signature (Fig. 5e–g).

## Discussion
The occurrence of drug treatment-associated CT, leading to decreased cardiac function, follows the therapeutic effects of the drugs, and is only observed in a subset of the patients using the drug. This raises the question of whether it would be possible to obtain early cell-based signatures predictive for drug toxicity. Here we addressed this question by attempting associating drug treatment-induced gene-expression patterns with the clinical risk for the adverse events of interest.

By estimating clinical risk from the FAERS database, our method utilizes a relevant and unbiased approach for the quantification of CT risk. As a result, our CT risk scores lack notable pitfalls such as selection bias associated with tightly controlled clinical trials, which underestimate adverse-event risks due to cohort size, trial duration, and selective inclusion criteria for subjects. Nevertheless, there are limitations to the FAERS database as well, which we have discussed and addressed in previous work[22]. Specifically, use of the FAERS resource may confound demographics information such as age and sex, which was

observed not to vary across different KIs. Moreover, CT risk score does not reflect absolute risk for developing CT. Rather, it reflects the relative risk for a subset of patients for which drug-associated adverse events were reported. In addition, there may be some systematic biases based on the sampling frequency of drugs by institution.

It remains unclear if all KIs induce CT through similar mechanisms, and to what extent ultimate clinical pathologies are similar. While the FAERS database allows us to distinguish between different types of CT, the annotation is not uniform and may either refer to distinct pathophysiological descriptions or rather more general clinical presentations of heart failure. To this end, we chose to lump all forms of heart failure, while excluding cardiac AEs that have known and unrelated origin such as coronary artery disease and arrhythmias.

We compared KI-associated transcriptomic response profiles generated from cultured human primary cardiomyocyte-like cells with clinical CT risk scores to obtain a reduced set of genes that may predict the relative risk for KI-associated CT. Using the clinically weighted signatures and the associated regression coefficients identified in the elastic net model, the relative risk for CT can be predicted. The risks predicted by our signatures and the associated regression model can be used in drug development to rank the potential risk of novel KIs with respect to existing KIs with better characterized clinical risks for CT.

The signatures generally showed good prediction of CT risk during cross-validation as well as on an independent set of KIs

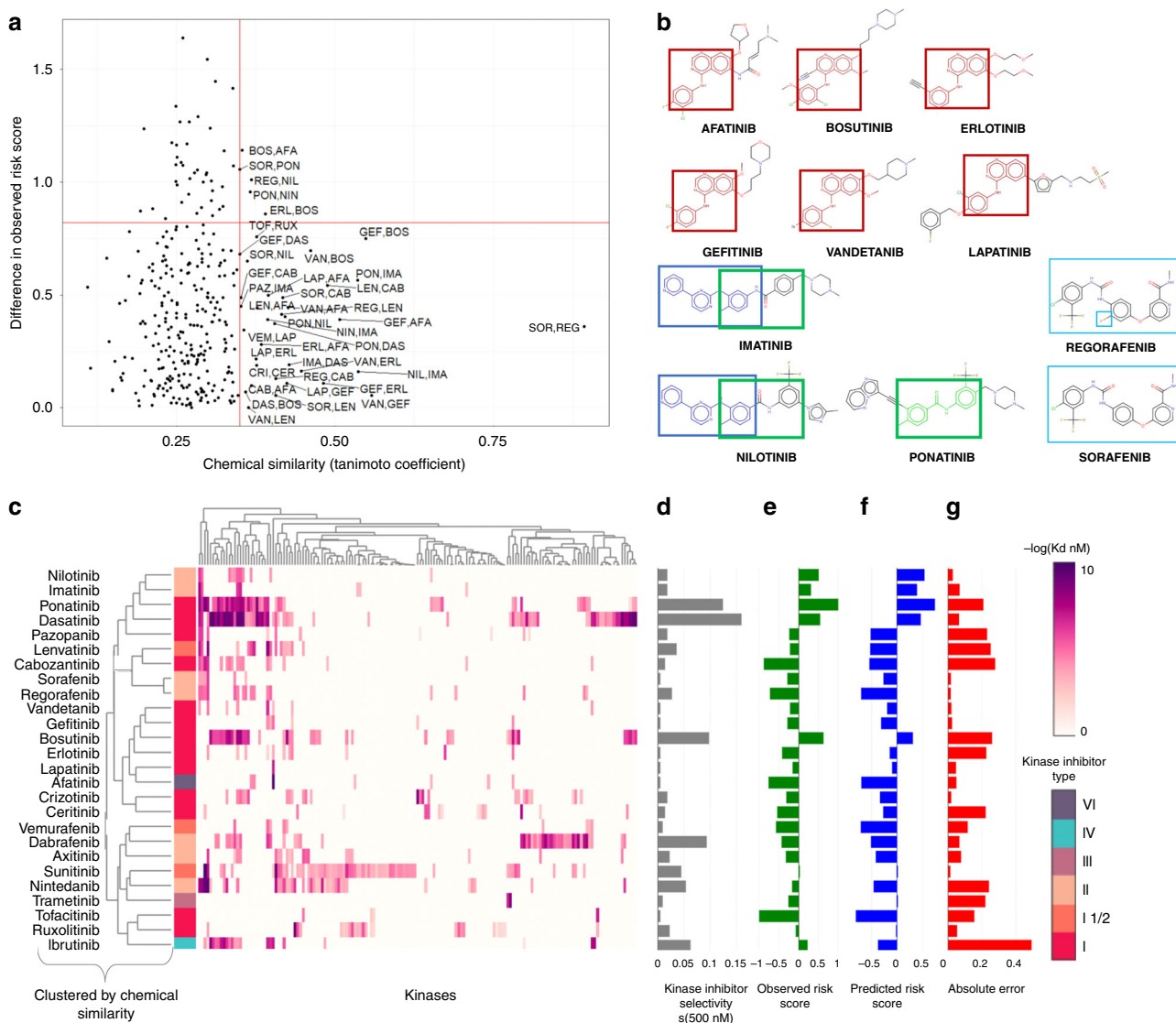

**Fig. 5 Structure–activity–similarity (SAS) maps of kinase inhibitor activity and cardiotoxicity. a** A SAS map relating pairwise chemical similarity measured by Tanimoto coefficient (Tc) derived from a weighted average of 4 chemical fingerprints (ECFP4, ECFP2, Daylight, and MACCS), between pairs of 26 kinase inhibitors (Table 1) and their difference in cardiotoxicity scores (DCS). The threshold for chemical similarity was the top 10% value in the distribution of Tc values: 0.38. The threshold value for DCS was half of the maximum DCS score: 0.82. **b** Highlighted chemical scaffolds for distinct kinase inhibitors observed in the upper- and lower-right regions. **c** Binding profile of kinase inhibitors based on data from Klaeger et al.[5]. Kinase inhibitors were hierarchically clustered based on chemical similarity, and kinase inhibitors are annotated by their binding mode (e.g., type I, type I1/2, type II, type III, type IV, or type VI)[6]. **d** Kinase inhibitor selectivity scores at 500 nM $K_d$. **e** Observed cardiotoxicity risk scores were normalized to zero and ordered based on hierarchical clustering of the kinase inhibitors. **f** Predicted cardiotoxicity risk scores were normalized to zero and ordered based on hierarchical clustering of the kinase inhibitors. **g** Absolute error from observed and predicted cardiotoxicity risk scores. Source data are provided in source data file.

(Fig. 4), while the only poorly performant KI, ibrutinib, inhibitor of Bruton nonreceptor protein-tyrosine kinase, represents a unique KI in terms of binding mode (i.e., type VI inhibitor) and high promiscuity (Fig. 5). Specifically, it is a member of an emerging class of kinase inhibitor drugs that bind their targets covalently (type VI KIs). These drugs are highly underrepresented in the databases used in this analysis, explaining the misclassification of ibrutinib[30].

The four cell lines we studied are insufficient to fully capture such human variability to KIs. Therefore, in our analysis, we used median fold-change gene-expression profiles across multiple cell lines. The resulting averaged gene-expression profiles thus reflect relatively consistent changes in gene expression across cell lines, i.e., changes in gene expression that are less likely to be highly

variable across cell lines, yet may also reflect a set of predictors that may be more consistent in the population. Given that the FAERS CT risk scores also reflect a population-level CT risk, the use of these median values in fold-change gene-expression values is a reasonable starting point for our analyses.

The experimental underpinning of the transcriptomic profiles generated in this study makes them likely to be of value in selecting drug candidates with a low risk for CT as an adverse event. Our analysis is based on primary human heart-derived cardiomyocyte-like cells. Although these cell lines do have phenotypic limitations due to dedifferentiation, the signatures obtained from the cells could be relevant for prediction of clinical drug effects. These cell lines may be reflective of human cardiac pharmacology, i.e., in comparison with animal-derived cardiomyocytes, even though further

**Table 1 Overview of KIs included in this analysis.**

| Drug | Three-letter code | Approval year[a] | Therapeutic targets | Concentration (μM)[b] |
|---|---|---|---|---|
| Afatinib | AFA | 2013 | ErbB2 and EGFR | 0.05 |
| Axitinib | AXI | 2012 | VEGFR1/VEGFR2/VEGFR3/PDGFRB/c-KIT | 0.2 |
| Bosutinib | BOS | 2012 | Bcr-Abl and SRC | 0.1 |
| Cabozantinib | CAB | 2012 | c-Met and VEGFR2 | 2 |
| Ceritinib | CER | 2014 | ALK | 1 |
| Crizotinib | CRI | 2011 | ALK and HGFR | 0.25 |
| Dabrafenib | DAB | 2013 | BRAF | 2.5 |
| Dasatinib | DAS | 2006 | ABL, ARG, KIT, PDGFRα/β, and SRC | 0.1 |
| Erlotinib | ERL | 2004 | ErbB1 | 3 |
| Gefitinib | GEF | 2003 | ErbB1 | 1 |
| Imatinib | IMA | 2001 | Bcr-Abl | 5 |
| Lapatinib | LAP | 2007 | ErbB1 | 2 |
| Nilotinib | NIL | 2007 | Bcr-Abl | 3 |
| Pazopanib | PAZ | 2009 | VEGFR2, PDGFRα/β, and KIT | 10 |
| Ponatinib | PON | 2012 | Bcr-Abl, BEGFR, PDGFR, FGFR, EPH, SRC, c-KIT, RET, TIE2, and FLT3 | 0.1 |
| Regorafenib | REG | 2012 | RET, VEGFR, and PDGFR | 1 |
| Ruxolitinib | RUX | 2011 | JAK | 1 |
| Sorafenib | SOR | 2005 | BRAF, VEGFRs, PDGFRα/β, FLT3, and KIT | 0.5 |
| Sunitinib | SUN | 2006 | VEGFR, PDGFR, CSF1R, FLT3, and KIT | 1 |
| Trametinib | TRA | 2013 | MEK1 and MEK2 | 0.1 |
| Tofacitinib | TOF | 2012 | JAK | 1 |
| Vandetanib | VAN | 2011 | RET, VEGFR, and EGFR | 0.33 |
| Vemurafenib | VEM | 2011 | BRAF | 2 |

[a]US approval date, first indication.
[b]Derived from maximum total (bound + free) plasma concentrations in humans as reported in the literature.
Table S3 lists the purity and literature references to clinical concentrations.

characterization and standardization are still needed. Detailed characterization of these cell lines is available as metadata to the RNAseq datasets at www.dtoxs.org. Our analyses used drug exposures similar to clinically reported maximum plasma concentrations of the individual KIs, rather than using the same concentrations for all KIs, even though we did not correct for protein binding. We expect that the duration of 48-h exposure may reflect transcriptomic changes that are likely related to early changes in subcellular processes associated with the adverse event of interest.

Unfortunately, in this study, it is not feasible or ethically possible, due to lack of prior informed consent, to compare cardiac gene-expression signatures with gene-expression profiles from patients who receive KI-therapy and/or who developed KI-associated CT. We considered whether we could compare our gene-expression signatures to cardiac gene-expression data from patients with heart failure who undergo surgery. Typically these are patients with advanced disease, and the gene expression in tissue from advanced disease is not likely to be of relevance to acute drug-induced CT.

By investigating the chemical structure and binding profile similarity of KIs, we are able to observe that chemical components and scaffolds that lead to promiscuous binding of KIs to multiple binding targets are correlated with higher CT values. This is consistent with the notion that a portion of CT risk of KIs can be attributed to higher levels of off-target interactions. Indeed, when we investigate the binding profile of three chemically similar KIs: afatinib, erlotinib, and gefitinib, we find that their binding profiles are fairly specific compared to other KIs, and they have a lower normalized CT risk score. One limitation we have observed with our approach is that chemically distinct KIs (e.g., in terms of binding profile, substructural similarity, and type), such as the type IV inhibitor ibrutinib, exhibit diminished predictive performance. However, we think that using the guidelines we provide herein, this signature could still assist in the development and prioritization of KIs with lower toxicity risks.

We cautiously anticipate that clinically weighted transcriptomic signatures such as those developed in this study may be of relevance to guide safety assessment in early drug development. Unlike the relatively well-established assessment of electrophysiological safety issues such as QT prolongation, the assessment of non-QT type of CT associated with KI[16] and other novel drugs[31], lacks reliable biomarkers. The transcriptomic signature for CT identified in this study may help fill this gap, especially if its structure and binding profiles are closely represented within the inhibitors in this study. One could anticipate that after initial selection of promising KIs with apparent efficacy in preclinical screens, transcriptomic profiling using the signatures developed here may possibly be used to rank drugs for the expected CT risk and exclude those with high CT risk scores (Supplementary Fig. 6).

While beyond the scope of this study, future extension of our studies could explore the idea of studying individualized risk scores for CT. That is, do baseline gene-expression profiles of larger libraries of patient-derived cardiomyocyte cell lines predict the difference in risk for CT between individual patients? Ideally, such an analysis would be conducted using induced pluripotent stem cell-derived cardiomyocytes from patients, who have received KIs and experienced different levels of CT, such as was recently described for anthracycline chemotherapeutics[32]. This would then further enable the development of precision medicine approaches to KI therapy that could minimize the risk for CT.

## Methods
**Cell culture and drug treatment**. Adult human cardiomyocytes (Cat #: C12810) were purchased from PromoCell GmbH (Heidelberg, Germany) and grown in culture as per the manufacturer's instructions. Four different cell line lots (Lot #: 3042901.2, 4031101.3, 2082801.2, and 2120301.2) isolated from two male and two female subjects were cultured under serum-free differentiation conditions for at least 28 days prior to drug treatment. Details regarding metadata information, including cell line metadata and the quality control and assurance metrics, can be found on www.dtoxs.org.

**Dose–response experiments**. For two of the four cell lines, dose–response experiments were conducted treating cells for 48 h with eight increasing perturbagen concentrations (5 nM, 50 nM, 100 nM, 500 nM, 1 μM, 5 μM, 10 μM, and 100 μM) and vehicle-treated control, in quadruplicates. We assayed for viability through image-based analysis of nuclear counts with Hoechst 33342 (Thermo Fisher, Cat #: H3570) and MitoTracker Red (Thermo Fisher, Cat #: M22426) for mitochondrial toxicity. Details of the experimental protocols for cell culture, drug treatment, and transcriptomics have been described as step-by-step standard operating procedures for the various experiments available on www.dtoxs.org.

**Transcriptomics**. Cells were treated for 48 h with a single perturbagen concentration around the maximal concentration (Supplementary Table 1). After drug treatment, the cells were lysed, RNA was collected using TRIzol, and gene-expression profiles were measured using the 3′ digital gene-expression method[33,34].

**Sequence alignment and processing of gene-expression data**. The raw sequences were demultiplexed. Combined standard RNAseq files were aligned to the reference human genome hg38 using the STAR software suite[35]. The resulting alignment files were parsed to identify the fragments with acceptable alignment quality, to remove duplicate fragments, and to assign accepted fragments to the corresponding genes. The resulting read-count (i.e., transcript count) table was then subjected to correlation analysis at each treatment condition, to identify and remove outlier samples, determined by predefined thresholds. The gene read-count tables were then subjected to differential gene-expression analysis using the R package EdgeR[36]. Details of these computational procedures are described elsewhere[23], and step-by-step protocols are available on www.dtoxs.org. The resulting normalized and log-transformed fold-change gene-expression values for each sample are also deposited for public access to the DToxS data repository (www.dtoxs.org).

**Processing and exploratory analysis of gene-expression data**. The median log-transformed gene-expression fold-change value was calculated across all cell lines for each individual KI. The resulting matrix of gene fold-change values by KIs was used for the regression analysis. To obtain insight into the general patterns present in this KI-perturbed transcriptomics dataset, we generated rankings of the top 500 genes for each drug, by their absolute mean fold-change value, i.e., whether positive or negative. For each of these KI-associated rankings, we determined the frequency of these changes being also present in the ranking of other drugs, e.g., the similarity in genes present in the top 250 gene lists for each KI. This was visualized using the Jaccard index, and by plotting the most highly drug-connected genes against the associated drugs. Principal component analysis for the first three principal components on the absolute mean fold-change values for each drug was performed to further assess similarity between drugs in their gene-expression values.

**Calculation of tissue cell line expression similarity**. Pairwise expression similarity scores were computed based on the Jaccard coefficient of a binary matrix based on RNA sequencing data from PromoCell cardiomyocyte exposures to kinase inhibitors. The top 500 genes for a KI were set as 1, while genes that were not in the top 500 were set as 0.

**Calculation of clinical risk RORs**. Adverse-event frequencies from the FDA Adverse Event Reporting System (FAERS) were obtained from the AERSmine resource[37], which contains a curated version of the FAERS database. ADRs in the FAERS database are organized according to MedDRA[38], which is a hierarchical ontology to classify ADRs from high-level organs associated with the pathology to reported low-level specific pathological conditions. We downloaded the frequencies of the occurrence of ADRs for all protein KIs available in FAERS, together with all other frequencies of ADRs reported for these KIs. A time-stamped record of this download to reproduce this analysis was retained. RORs were then computed for each KI using the frequency $f_{dt}$ of the ADR of interest, the frequency $f_{dn}$ of any other ADR occurring, the frequencies $f_{nt}$ of occurrence of the ADR of interest for any other protein kinase inhibitor, and the frequency $f_{nn}$ for all other ADRs and KIs. The ROR was calculated using Eq. (1)

$$ROR = \frac{f_{dt}/f_{dn}}{f_{nt}/f_{nn}}, \qquad (1)$$

whereas the standard error (SE) of the log ROR was calculated using Eq. (2)

$$SE_{logROR} = \sqrt{\frac{1}{f_{dt}} + \frac{1}{f_{dn}} + \frac{1}{f_{nt}} + \frac{1}{f_{nn}}}, \qquad (2)$$

with the log-transformed confidence interval (CI) being calculated as follows: $CI = log(ROR) \pm 1.96 * SE^{logROR}$.

Adverse events in FAERS are mapped to the MEDDRA dictionary[38]. CT events related to heart failures and cardiomyopathies, excluding arrhythmogenic ADRs and coronary artery disorders, were selected from the main MEDDRA cardiac ADR group. The selected ADRs primarily reflected different stages of heart failure, which were grouped together.

**Elastic net regression analysis**. The FAERS-derived risk RORs for CT were regressed against the KI-associated vectors of mean fold-change values across the four cell lines. A two-step regression procedure was then used to select predictor genes reducing the sensitivity to changes in dataset composition. For this, we first generated 1000 bootstrap datasets with replacements for gene expression–KI risk score pairs. Each of these bootstrap datasets was fit using an elastic net regression model (R version 3.4.3, package glmnet, version 2.0-16). The genes that were selected as predictors (i.e., nonzero regression coefficient) and the scaled values of the gene-associated coefficients were saved for each bootstrap dataset. Across all bootstrap datasets, the relative frequency of the selection of gene-based predictors, and the mean-scaled coefficient value was computed. We then calculated the product of the mean frequency and scaled coefficient value, rank predictors by their importance with respect to robustness (selection frequency). A large number of percentiles of these rankings were evaluated using leave-one-out cross-validation. The selection percentile (99.755%) resulting in optimal prediction errors (RMSE) was then used to select a subset of gene-based predictors, and the model that generated the final gene-expression signatures. The selected predictor genes were then ranked by their relative importance, and by their median fold-change values, and displayed as clustered heatmaps. We finally evaluated the predictive value of the resulting regression model to predict CT risk scores for the two left-out KIs.

When using this approach to analyze similar datasets of cardiomyocyte transcriptomes together with risk scores, it is possible that potentially different genes are identified than those described in the current report. This difference associated with the intrinsic property of penalized regression approaches that select predictors from potentially highly correlated sets of predictor candidates. Hence, small changes in either risk scores or gene-expression datasets may affect correlation structures of the data and thereby the list of genes for a signature.

**Enrichment and network analyses**. Enrichment analysis was performed based on a one-tailed Fisher's exact test using R (package stats), in order to identify enrichment of specific genes in predefined gene lists. For enrichment of pathways and biological processes, we used the KEGG database (2016), and for enrichment of protein kinases, we used the KEA database (2015). Diseases were excluded from the KEGG list of processes (e.g., diabetes, depression, and cancer), in order to only evaluate general biological processes or pathways. We used the top 250 DEGs ranked by $p$ value for each KI to perform enrichment analysis. Subsequently enriched term $p$ values were correlated with CT risk scores to identify kinases and pathways associated with CT risk.

The gene part of the signature for CT identified in the regression analysis was used as seed note to perform a protein–protein interaction network (PPI) analysis, conducted using the web application X2K[39], which aims to identify associated kinases and transcription factors based on multiple PPI databases.

**Calculation of chemical similarity**. RDkit (www.rdkit.org)[40] was used to generate chemical fingerprints and compute pairwise Tanimoto coefficients (Tc) between the 26 tested kinase inhibitors. For each pair of inhibitors, we first calculated the Tc using four chemical fingerprints, including Morgan_2 2,048-bit (ECFP4)[41], Morgan_1 2,048-bit (ECFP2)[41], Daylight-like[42], and MACCS[43]. Because each of these fingerprints capture distinct chemical properties, we computed a weighted Tc average of the three fingerprints: 30% ECFP4, 30% ECFP2, 30% Daylight-like, and 10% MACCS, which exhibited the most optimal spread of the distribution of the pairwise distances. To generate the SAS maps (Fig. 5a), we plotted the pairwise-weighted Tc values with their difference in CT scores (DCT). Finally, 0.35 was set as the threshold for chemical similarity, while half of the maximum difference was set as the threshold for DCS. Chemical structures were drawn using Marvin (www.chemaxon.com)[44] based on SMILES strings obtained from PubChem.

**Calculation of KI-binding target similarity**. Kinome-wide kinase inhibitor-binding ($K_d$) profiling data were obtained from Klaegar et al.[5], which consisted of kinome-binding (Kd) profiling data for all of the tested kinase inhibitors across 242 kinases. A heatmap was generated for selected kinase inhibitors based on the negative log of the $K_d$ values from Klaeger et al. (Fig. 5c)[5]. Notably, the $K_d$ values were scaled by 100,000 to avoid negative log values.

**Reporting summary**. Further information on research design is available in the Nature Research Reporting Summary linked to this article.

## Data availability

All processed RNAseq data and the curated version-controlled standard operating procedures featured in this study can be downloaded freely at (www.dtoxs.org)[22] or the LINCS Data Portal (http://lincsportal.ccs.miami.edu/dcic-portal/). Raw transcriptomics data can be accessed through the Gene Expression Omnibus (GEO) repository with accession numbers GSE146096 and GSE146097. Source data for each figure are provided with this paper. All remaining data will be available from the corresponding author upon reasonable request. Source data are provided with this paper.

## Code availability

All scripts are open-source and available from the DToxS GitHub repository (https://github.com/dtoxs).

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

## Acknowledgements

This project was supported in part by the NIH LINCS center grant (U54 HG008098) and the Systems Biology Center grant (P50 GM071558). J.G.C.H. received funding from the European Union MSCA program (Project ID 661588). This work was partially carried out using the Dutch national e-infrastructure with the support of SURF Foundation.

## Author contributions

J.G.C.H. and R.R. performed the data analysis; J.G.C.H., R.R., J.H., M.R.B., E.S., A. Sc., E.U.A., and R.I. wrote the paper; Y.X. performed RNAseq data processing; A.P. and J.M.G. performed the mass spectrometry drug purity analyses; A.St., B.H., G.J., and J.V.S. performed the cell culture, drug perturbation, and the RNA isolation; E.U.A. supervised the experimental efforts; E.U.A. and J.M.G. determined the experimental drug concentrations and purity; M.M. supervised the RNA sequencing; J.G. supervised the quality assurance and assay reproducibility; A.Sc. supervised the cheminformatics analysis; R.I. conceived the project; all authors reviewed the paper.

## Competing interests

R.R. and A.S. are co-founders of Aichemy Inc. The remaining authors declare no competing interests.
