## [Peer Review File · Nature Communications]

Reviewers' comments:

Reviewer #1 (Remarks to the Author):

van Hasselt et al. reported a new discovery strategy and initial results that address a relevant, timely challenge: the prediction of cardiotoxicity associated with kinase inhibitors. Although the findings are encouraging, the supporting evidence is not sufficiently strong. Overall, there is a need for further research to demonstrate the potential relevance of the proposed models for either patient management or drug development applications.

Apart from the limited sample sizes available for generating and testing models, there is a need for analyzing independent data sets for reproducing models or evaluating their predictive capability. Ideally, this should include independent patient-derived gene expression and adverse events data sets, or at least the former.

The study does not report the drug toxicities, in vitro, for the different cell lines investigated. Additionally, their treatment used typical drug concentrations as reported in the literature. It would be important to know the actual effect of different doses on specific cell lines. A key question, for example, is whether the selected doses and treatment durations can induce early, non-toxic effects in vitro.

Although the study focuses on the investigation of post-treatment transcriptomic signatures, I wonder if prediction models using untreated cell lines were investigated. This also represents a clinically-relevant application scenario. At least, it is important to investigate correlations between (baseline) gene expression patterns and risk, as well as the differences between these expression patterns and those observed after treatment.

Genes without biological function annotations were excluded from the regression analyses. The rationale for this gene filtering deserves elaboration. In the final selection of genes (after bootstrap): "multiple upper percentiles of these rankings were regressed", and the one giving the lowest error was selected. Such percentile values are missing from the main text.

After selecting models with all the data available, a new model was fitted and tested via cross-validation with four drugs out. In comparison to the full-data model: How robust was such gene selection and estimation of coefficients? In any case, evaluations using independent data would be required.

The article requires deeper insights into the molecular mechanisms underlying the different relative risks, drugs or adverse events. The article offers a basic description based on a standard pathway annotation analysis.

Although the authors indicated that the predicted scores are not “absolute” risk scores for specific patient-derived biological samples, and that rather they represent “relative” risk for sets of patients with reported events, this distinction was not sufficiently emphasized in the article. Different parts of the article may actually give the impression that the authors claim that the models are relevant for estimating patient-specific risk.

The lists of signature genes and their coefficients should be made available as supplementary text, a supplementary figure is not sufficient. More details for the implementation of the selected regression models is needed. In general, the provided information makes it difficult to reproduce (or simply verify) the reported models.

Reviewer #2 (Remarks to the Author):

This is a comprehensive "top down" analysis of tyrosine kinase inhibitor drugs and the prediction of their likelihood of causing congestive heart failure. There are a few comments.

1. Separating degree of heart failure by MedDRA terms is not likely very valid.
2. There are a number of typos that should be corrected.

Reviewer #3 (Remarks to the Author):

In this article, Hasselt et al. obtained the transcriptomic data of four normal human heart derived primary cardiomyocyte cell lines that were treated with various kinase inhibitors and proposed a cell-based model to use these transcriptomic signatures to predict the drug-induced cardiotoxicity. The authors conclude that proposed model can provide predictive signatures for drug-induced cardiotoxicity that can be used for drug development and precision medicine. The subject is very

relevant, proposed model is novel, these models are warranted from a long time and authors conclusion seems reasonable. However, I have a major concern that needs to be addressed.

1. My only major concern is the proposed model and conclusion needs to be validated with some in-vivo data of comparative toxicity with +tive and –controls. There are no reports of cardiac events with many drugs with higher Z-score in Fig 2B, basically questioning the predictive value of the proposed model. For example, ponatinib has the highest Z score for ventricular dysfunction (Fig 2B). However, in the clinical setting, most of the ponatinib adverse effects are limited to vascular toxicity only (Thrombosis and Atherosclerosis). Thus, the proposed model could be misleading as well and the predictive value of the model needs to be supported with some in-vivo data. The Very similar statement can be made for several other small molecule inhibitors with highest Z score for HF phenotype.

2. One minor point is, nintedanib and ibrutinib are included in Fig 2B but missing from Table 1.

Response to reviewer comments

Reviewer #1 (Remarks to the Author):

van Hasselt et al. reported a new discovery strategy and initial results that address a relevant, timely challenge: the prediction of cardiotoxicity associated with kinase inhibitors. Although the findings are encouraging, the supporting evidence is not sufficiently strong. Overall, there is a need for further research to demonstrate the potential relevance of the proposed models for either patient management or drug development applications.

We appreciate the reviewer's positive assessment of our work. Previously we suggested that our work could be of relevance both to drug development and patient care. However, we have now revised our manuscript to focus entirely on application predicting cardiotoxicity risk of kinase inhibitors in a drug development context, because we think this is the primary application of our work. To further address the potential relevance we have made the following changes:

- 1. We provide a specific discussion and flowchart figure that proposes a workflow for integrating the developed signatures in a drug development trajectory (Figure 6).**
- 2. We have added a comprehensive table (Figure 1D) which summarizes current available in vitro and in vivo phenotypic assays that could be used in drug development to study cardiotoxicity,. These data from the literature and our own data show that KI-induced cardiotoxicity is poorly predictable using such cell or whole animal based assays.**
- 3. We now perform an updated signature analysis that leaves out a low and high risk cardiotoxicity drug entirely, confirming the predictive relevance for drug development of new PKIs in contrast to the previous analysis that used all available KIs.**
- 4. We have performed an analysis that relates chemical substructure of KIs to cardiotoxicity risk scores which shows potential to perform structure based optimization of high risk drugs.**
- 5. After acceptance all our scripts will be uploaded to the DTOXS Github repository, which will include a user-friendly script for prediction of cardiotoxicity risk scores that can be used in a drug development context.**

Apart from the limited sample sizes available for generating and testing models, there is a need for analyzing independent data sets for reproducing models or evaluating their predictive capability. Ideally, this should include independent patient-derived gene expression and adverse events data sets, or at least the former.

We fully agree and appreciate the need for additional independent datasets.

As suggested by the reviewer we have now included another adverse event dataset from WHO which confirms the risk scores obtained from FAERS. The WHO dataset contains adverse event data from non-US sources. When comparing FAERS and WHO risk scores, similar risk scores were found (Figure 1C).

Although we agree that patient-derived genes expression would be highly valuable to validate our signatures. This would thus require us to obtain heart biopsy data from patients who receive KI-therapy and/or who developed KI-associated cardiotoxicity. We are convinced this is not ethically possible and believe this is therefore not a viable option to include. We did consider the possibility of comparing existing gene expression data from patients with general heart failure. However, we think this will not be of relevance to acute drug-induced cardiotoxicity, as heart failure often takes many years to manifest itself before surgery and patients who undergo surgery may be at advanced stage of the disease.

We have added a comprehensive Figure 1D, which summarizes currently available in vitro and in vivo assays, which shows a lack of predictive assays for PKI-induced cardiotoxicity in drug development. From this literature overview it can be concluded that currently no acceptable in vitro or in vivo assays for cardiotoxicity induced by KIs exist. This was also confirmed by experts in the field of KI-induced cardiotoxicity (*Force, T. & Kolaja, K. L. Cardiotoxicity of kinase inhibitors: the prediction and translation of preclinical models to clinical outcomes. Nat. Rev. Drug Discov. 10, 111–26*). From this we can conclude that conducting additional in vivo experiments to confirm cardiotoxicity would not be meaningful. In fact, if such in vivo or in vitro assays would be predictive, no need would exist for the transcriptomic predictive signatures developed in our study, which are directly based on patient-reports for KI-associated cardiotoxicity.

The study does not report the drug toxicities, in vitro, for the different cell lines investigated. Additionally, their treatment used typical drug concentrations as reported in the literature. It would be important to know the actual effect of different doses on specific cell lines.

We previously choose not to report these because of known lack of predictive relevance for measuring drug toxicity phenotypes (e.g. viability or stress markers). It has been suggested that that KI-induced cardiotoxicity occurs actually at concentrations lower than those were viability decreases. However as requested we have now performed an extensive set of additional viability and mitochondrial stress dose-response experiments on the PromoCell cardiomyocytes (Figure 1E, Figures S1), that confirm that these assays are not directly predictive of CT, as expected. This finding only further supports the need for transcriptomic cardiotoxicity signatures described in this manuscript.

A key question, for example, is whether the selected doses and treatment durations can induce early, non-toxic effects in vitro.

The *in vitro* experiments measuring mitochondrial stress and viability show that at the clinical concentrations there is negligible cytotoxicity or mitochondrial stress for almost all low- medium and high risk drugs. This suggests that effects of KIs on cardiomyocytes are more subtle. Using transcriptomic profiling conducted we show the nature of KI-induced changes in gene expression that could provide insight into mechanisms, and predict, KI-induced cardiotoxicity.

Although the study focuses on the investigation of post-treatment transcriptomic signatures, I wonder if prediction models using untreated cell lines were investigated. This also represents a clinically-relevant application scenario. At least, it is important to investigate correlations between (baseline) gene expression patterns and risk, as well as the differences between these expression patterns and those observed after treatment.

Our analysis uses differential gene expression for the signatures compared to untreated cell lines to assess changes in gene expression after exposure to KIs.

We agree with the reviewer that studying baseline gene expression in cardiomyocytes to determine risk factors for cardiotoxicity is of great interest to estimate individual risk of patients. However we believe such an analysis should be conducted on a large amount of different cell lines. Using the four cell lines used in the current study, such an analysis would not be meaningful. In addition, we have now focused the application of this work on the urgent need in drug development to identify drug candidates where this is an overall increased risk for occurrence of cardiotoxicity.

We have however added this important suggestion of the reviewer to our discussion (last paragraph).

Genes without biological function annotations were excluded from the regression analyses. The rationale for this gene filtering deserves elaboration.

We removed the filtering step that excluded genes based on whether they were biologically annotated. Moreover, the reference genome used for annotation was updated to a more recent version which led to many more genes being adequately named.

In the final selection of genes (after bootstrap): “multiple upper percentiles of these rankings were regressed”, and the one giving the lowest error was selected. Such percentile values are missing from the main text.

We have added this value to the methods section.

After selecting models with all the data available, a new model was fitted and tested via cross-validation with four drugs out. In comparison to the full-data model: How robust was such gene selection and estimation of coefficients? In any case, evaluations using independent data would be required.

Previously we fitted both a full-data model and a reduced data model leaving two drugs out for each of the two CT subtypes. We decided, based on Reviewer2’s comments to group all CT subtypes into one category because it became apparent to us that these subtypes were not comparable with each other. Rather now, we focused on all forms of cardiac function loss and heart failure, only excluding electrophysiological side effects.

We agree that evaluation using entirely independent data would strengthen the value of our signatures, and this is how we have now implemented the analysis. Two KIs (low and high risk) were left out entirely from the very start of the analysis. We then predicted for these two independent drugs their clinical risk scores successfully using the developed signature (Figure 4D). We chose to only exclude two KIs given that overall only 23 KIs were available. We believe that using a larger test set would impact the generation of the signature to strongly.

In addition, we have now added standard deviations of predictions to show the variability obtained after repeated cross validation, to provide further insight into the robustness of the regression model.

The article requires deeper insights into the molecular mechanisms underlying the different relative risks, drugs or adverse events. The article offers a basic description based on a standard pathway annotation analysis.

We have now conducted a correlational analysis of both enriched kinases and enriched pathways (KEGG) in relation to clinical CT risk scores, identifying a number of kinases and pathways that showed strong negative or positive correlation with CT risk scores (Figure 3). In addition we have now conducted a protein-protein interaction (PPI) network analysis based on the signature genes that led to the identification of a PPI network identifying potential transcription factors and kinases associated with the predictive signature. Finally we have added an extensive figure that shows all enriched kinases and processes based on each of the 23 KI-perturbation transcriptomic profiles. However, as stated in the introduction, our goal was to provide a top-down approach surveying many KIs for their effects on gene expression. Detailed mechanistic studies are beyond the scope of the current contribution, and will be addressed in future contributions.

Although the authors indicated that the predicted scores are not “absolute” risk scores for specific patient-derived biological samples, and that rather they represent “relative” risk for sets of patients with reported events, this distinction was not sufficiently emphasized in the article. Different parts of the article may actually give the impression that the authors claim that the models are relevant for estimating patient-specific risk.

We have now removed the sentences that may suggest that patient-specific risk can be predicted. Rather, relative cardiotoxicity risk scores are predicted that provide insight into how different KIs have different relative risks for CT to occur in a population of patients. Moreover we have added Figure 1A to further explain how the risk scores are derived, and how they should be related back to overall drug usage in the patient population.

The lists of signature genes and their coefficients should be made available as supplementary text, a supplementary figure is not sufficient. More details for the implementation of the selected regression models is needed. In general, the provided information makes it difficult to reproduce (or simply verify) the reported models.

Signature genes and regression coefficients are now added as Supplemental Table S3. After acceptance of the manuscript we will also upload an R script to show full reproducibility, and, a user-friendly script to directly apply the model to transcriptomic datasets.

Reviewer #2 (Remarks to the Author):

This is a comprehensive "top down" analysis of tyrosine kinase inhibitor drugs and the prediction of their likelihood of causing congestive heart failure. There are a few comments.

1. Separating degree of heart failure by MedDRA terms is not likely very valid.
2. There are a number of typos that should be corrected.

As suggested by the reviewer we have now pooled all heart failure related terms into one risk group excluding arrhythmia related cardiac events. This reduced the assumptions made about the underlying nature of the pathophysiology.

We have thoroughly reviewed the paper for typos and corrected these. We apologize for these errors

Reviewer #3 (Remarks to the Author):

In this article, Hasselt et al. obtained the transcriptomic data of four normal human heart derived primary cardiomyocyte cell lines that were treated with various kinase inhibitors and proposed a cell-based model to use these transcriptomic signatures to predict the drug-induced cardiotoxicity. The authors conclude that proposed model can provide predictive signatures for drug-induced cardiotoxicity that can be used for drug development and precision medicine. The subject is very relevant, proposed model is novel, these models are warranted from a long time and authors conclusion seems reasonable.

However, I have a major concern that needs to be addressed.

1. My only major concern is the proposed model and conclusion needs to be validated with some in-vivo data of comparative toxicity with +tive and –controls.

While we fully appreciate the concern of the reviewer regarding in vivo validation we believe this is not possible currently for kinase inhibitors. We have added an extensive overview of published in vitro and in vivo experiments to study KI-associated cardiotoxicity phenotypes. The conclusion from this overview, and also from experts in the field as previously mentioned, is that no reliable in vitro or in vivo models for this type of cardiotoxicity are currently available. We have confirmed this by conducting new experiments with the PromoCell cardiomyocytes used for this project, which show the same lack of correlation of phenotypic cellular toxicity and risk for cardiotoxicity. In addition we would like to stress that if such experiments would have been possible, the need for the predictive transcriptomic cardiotoxicity signature would not exist anymore.

There are no reports of cardiac events with many drugs with higher Z-score in Fig 2B, basically questioning the predictive value of the proposed model. For example, ponatinib has the highest Z score

for ventricular dysfunction (Fig 2B). However, in the clinical setting, most of the ponatinib adverse effects are limited to vascular toxicity only (Thrombosis and Atherosclerosis). Thus, the proposed model could be misleading as well and the predictive value of the model needs to be supported with some in-vivo data. The Very similar statement can be made for several other small molecule inhibitors with highest Z score for HF phenotype.

We respectfully disagree with this comment. It is selective and not fully supported by the data. We did not claim that cardiotoxicity is the most prominent adverse drug effect for any of the KIs for drug ranked high for CT risk. KIs typically are associated with multiple adverse effects. However, we believe that cardiotoxicity , resulting in decrease in cardiac function is of particular concern considering the long-term effects that may have for patients. We would like to stress that individual patient-reports of adverse drug effects in the FAERS database (over 10 million reports currently) is a highly relevant resource to identify drugs associated with particular increased risks for side effect profiles in humans under standard clinical testing. Pharmacovigilance databases such as FAERS have been designed for this very purpose by drug regulatory authorities. In contrast, clinical case reports and similar publications are based on much fewer patients. In addition it has been suggested by clinical cardio-oncology experts that KI-induced cardiotoxicity remains often missed as cardiac monitoring is not regularly conducted for many KIs. We believe that the risk scores derived using the FAERS database in this publication are of significant relevance. Of course there do exist limitations to FAERS-derived risk scores as well, and we have expanded the discussion about this topic.

For ponatinib, which is mentioned by the reviewer, it has been reported from meta-analysis of three clinical trials that cardiac failure and cardiovascular events are frequently occurring adverse events (Dorer DJ et al Leuk Res. 2016 Sep;48:84-91) besides other adverse effects such as hematological toxicity, vascular effects and others. Several other studies have confirmed this.

In vivo and in vitro phenotypic studies for KI-induced cardiotoxicity are not predictive and cannot reproduce cardiotoxicity for agents where the occurrence of cardiotoxicity is widely accepted (such as trastuzumab, imatinib and others); this is a major challenge in drug development of new KIs as it limits preclinical screening for this side effects. We have summarized this point in Figure 1D-E, as previously discussed in answers to other reviewers. We have however now fully excluded two KIs (low and high CT risk) from our signature development procedure, and for these KIs we have shown that the signature can predict CT risk well.

2. One minor point is, nintedanib and ibrutinib are included in Fig 2B but missing from Table 1. Thank you for pointing this out. We have corrected this error. These two drugs should not have been included in the figure and have now been removed.

Reviewers' comments:

Reviewer #1 (Remarks to the Author):

The authors addressed all my comments. However, although they justified why they could not do more regarding their independent testing, such an evaluation is still limited and is not sufficient to support strong conclusive claims.

Reviewer #3 (Remarks to the Author):

In this article, Hasselt et al. obtained the transcriptomic data of four normal human heart derived primary cardiomyocyte cell lines that were treated with various kinase inhibitors and proposed a cell-based model to use these transcriptomic signatures to predict the drug-induced cardiotoxicity. The authors conclude that the proposed model can provide predictive signatures for drug-induced cardiotoxicity that can be used for drug development. Of note, in the original submission authors also advocated for precision medicine as a key application of the proposed model. It's a good design study and nicely presented. I have following comments.

1. Since there is a long gap between original submission and the revision version (about 2 years), this manuscript this factual update, e.g. The second line of introduction states "There are currently more than 28 KIs approved for clinical use by the FDA, it was may be true in early 2017, however, currently this list goes to around 40 KIs. Authors need to update these facts with the current status.
2. Authors will be wondering what the basis was to choose 23 compounds out of currently approved 40 compounds. I could not find a rationale in the current manuscript for the same.
3. This reviewer still feels that the scope of transcriptomic profiling of just 4 cell lines to predict the overall cardiotoxicity of all TKIs is limited and needs further validation with additional models.
4. Line 79, "Reference no 1920" should be 19, 20 (comma missing)
5. Line 185 spaces missing in "of26geneswith"

Reviewer #1 (Remarks to the Author):

The authors addressed all my comments. However, although they justified why they could not do more regarding their independent testing, such an evaluation is still limited and is not sufficient to support strong conclusive claims.

We agree that without independent testing, our model did not merit conclusive and widespread claims. To address this, we actually performed a new independent set of experiments that covered three additional KIs (previously not included in our study). We observed agreement with our transcriptomic signature predictions and the observed relative risk scores derived from FAERS. (See *Identification of a transcriptomic signature to predict cardiotoxicity risk*, and Figure 4D). Importantly, the independent set of experiments were subjected to a different experimental transcriptomic method than the previous KIs, demonstrating the robustness of our signature to distinct transcriptomic approaches. We think that the additional work conducted, based on the constructive remarks of our reviewers, make the predictive capabilities of our model more convincing. We thank the reviewer for their valuable suggestions.

Next, we also have new additional computational analyses that are based on drug structures and activity profiles (Figure 5). We expanded our SAS map from the previous submission to include the 3 additional kinase inhibitors tested. Next we utilize a newer and significantly larger kinome-wide binding profile dataset from Klaegar et. al 2017 (Figure 5C) to elucidate the relationship between binding profile and kinase inhibitor selectivity (Figure 5D), observed (Figure 5E) and predicted (Figure 5F) risk scores and the absolute error between observed and predicted relative risk scores (Figure 5G). Finally we link binding profile with structural information by clustering KI based on chemical similarity and by kinase inhibitor binding mode. Unlike the Davis et al. set used in the previous submission, this data includes the binding profile of all 26 of the tested kinase inhibitors and thus is a better analysis.

Finally, we have toned down the definitive statements in our discussion and included two new segments that discuss the limitations of our approach.

Reviewer #3 (Remarks to the Author):

In this article, Hasselt et al. obtained the transcriptomic data of four normal human heart derived primary cardiomyocyte cell lines that were treated with various kinase inhibitors and proposed a cell-based model to use these transcriptomic signatures to predict the drug-induced cardiotoxicity. The authors conclude that the proposed model can provide predictive signatures for drug-induced cardiotoxicity that can be used for drug development. Of note, in the original submission authors also advocated for precision medicine as a key application of the proposed model. It's a good design study and nicely presented. I have following comments.

We thank the reviewer for their kind comments and constructive criticism.

1. Since there is a long gap between original submission and the revision version (about 2 years), this manuscript this factual update, e.g. The second line of introduction states "There are currently more than 28 KIs approved for clinical use by the FDA, it was may be true in early 2017, however, currently this list goes to around 40 KIs. Authors need to update these facts with the current status.

We realize that numerous additional KIs are now FDA approved (four more were approved just last week). We updated our introduction and updated our citations accordingly.

2. Authors will be wondering what the basis was to choose 23 compounds out of currently approved 40 compounds. I could not find a rationale in the current manuscript for the same.

We had begun the study with all FDA approved KIs at the time of study initiation about four years ago. As the reviewer points out, we could have expanded the study to more drugs as they were being approved midway; however, we note that our model development does not only depend on FDA approval, but more critically the reporting of adverse events on the FAERS database. At the time of the conceptualization of this study, we saw that these 23 KIs had reasonable amounts of FAERS data on their cardiotoxicity (i.e., they have been in use the longest time period). We now include this point to the introduction and to the discussion (as a limitation of our approach).

3. This reviewer still feels that the scope of transcriptomic profiling of just 4 cell lines to predict the overall cardiotoxicity of all TKIs is limited and needs further validation with additional models.

Our approach focuses only on the identification of targetable mechanisms that will likely be abundant in cardiomyocytes. We explicitly do not aim to predict inter-individual variation, but rather see our approach as a way to de-risk development of novel KIs in drug development, where extensive panels of cell lines or patient-derived cells would not be practical or accessible. We now include this rationale in our discussion and introduction. We also expanded our drug structural analyses section to further emphasize this point. We also elaborate in the discussion that while our specific transcriptomic signatures are limited to the 26 KI tested in this study, that our platform can be extended to all clinically used KIs to estimate potential cardiotoxicity

That being said, we concede that our transcriptomic profiling approach without *independent* validation falls short. To address this comment, we have performed additional transcriptomic experiments with all lines using three new KIs that were previously untested using a completely different sequencing approach (see *Identification of a transcriptomic signature to predict cardiotoxicity risk*, and Figure 4D). In order to control for the effects of assay diversity, we also included one of the older KIs. Our results show that the four independently tested drugs have similar predictive performance to the two internal validation samples. To elucidate why certain KI exhibit higher observed risk scores, we expanded our drug structure analysis, which now includes insights not only on the associations of substructures with cardiotoxicity, but also the

binding profile of the KIs as well as the binding mode of these KIs (figure 5). We show that certain chemically similar KI have similar cardiotoxicity risk scores, which are consistent with both their kinome-wide binding profile (i.e. binding to off-target kinases) as well as with their binding mode. For example, kinase inhibitors ponatinib and dasatinib are observed to have higher observed risk scores, are chemically similar, have similar binding mode (both are 'type I' KIs) and are both promiscuous kinase inhibitors, with many off-target interactions. Accordingly, we have remodeled our discussion to include these new findings as well as an expanded examination of the limitations of our approach. We thank the reviewer for noting these shortcomings and considerably improving our study.

4. Line 79, "Reference no 1920" should be 19, 20 (comma missing)

This has been fixed.

5. Line 185 spaces missing in "of26geneswith"

This has been fixed. We apologize for the oversight.

REVIEWERS' COMMENTS:

Reviewer #1 (Remarks to the Author):

The authors have adequately addressed my concerns. In particular, they included new analyses for an independent set of KIs. The revised version of the article also further clarifies key limitations of the study.

Reviewer #4 (Remarks to the Author):

The structure-activity considerations reported in the revised manuscript including the SAS map analysis presented in Figure 5A (it should read "Tanimoto") are reasonable and do not require further revisions.

Reviewer #1 (Remarks to the Author):

The authors have adequately addressed my concerns. In particular, they included new analyses for an independent set of KIs. The revised version of the article also further clarifies key limitations of the study.

We thank the reviewer for their positive evaluation of our manuscript.

Reviewer #4 (Remarks to the Author):

The structure-activity considerations reported in the revised manuscript including the SAS map analysis presented in Figure 5A (it should read "Tanimoto") are reasonable and do not require further revisions.

We fixed the capitalization of Tanimoto in Figure 5A and thank the reviewer for their positive evaluation of our manuscript.